# Chemical Composition, Fatty Acid Profile, and Lipid Quality Indices in Commercial Ripening of Cow Cheeses from Different Seasons

**DOI:** 10.3390/ani12020198

**Published:** 2022-01-14

**Authors:** Beata Paszczyk, Magdalena Polak-Śliwińska, Anna E. Zielak-Steciwko

**Affiliations:** 1Department of Commodity and Food Analysis, Faculty of Food Sciences, University of Warmia and Mazury in Olsztyn, 10-726 Olsztyn, Poland; m.polak@uwm.edu.pl; 2Department of Cattle Breeding and Milk Production, Wroclaw University of Environmental and Life, 51-630 Wrocław, Poland; anna.zielak-steciwko@upwr.edu.pl

**Keywords:** fat, protein, fatty acids, CLA, *trans* isomers, lipid quality indices, cow cheese

## Abstract

**Simple Summary:**

Cheese and other dairy products are important components of the diet that have a positive effect on human health. These products include substantial amounts of important nutrients, including proteins, bioactive peptides, amino acids, fat, fatty acids, and vitamins. The quantitative composition of fatty acids in milk fat changes under the influence of various factors, such as animal feeding, breed, lactation stage, individual characteristics, climatic conditions, health, age, and others. Of the above-mentioned factors, the most important influence is the diet. This work presents chemical composition and fatty acid profile, with particular emphasis on *trans* isomers (*cis*9*trans*11 C18:2 (CLA), C18:1 and C18:2 isomers) and lipid quality indices in ripening of cow cheeses from different seasons. The first batch contained cheeses produced in winter and purchased from the market between May and June 2020. The second batch contained cheeses produced in summer and purchased from the market between November and December 2020. Results obtained show differences between summer and winter cheeses in their chemical composition, the content of lipid quality indices, and fatty acids.

**Abstract:**

The aim of the study was to compare and demonstrate whether commercial rennet ripening cheeses available on the market in summer and winter differ in their chemical composition, fatty acid profile, content of *cis*9*trans*11 C18:2 (CLA) acid and other *trans* isomers of C18:1 and C18:2 acid and whether they are characterized by different values of lipid quality assessment indices. The experimental material consisted of rennet ripening of cheeses produced from cow’s milk available in the Polish market. The first batch contained cheeses produced in winter and purchased from the market between May and June. The second batch contained cheeses produced in summer and purchased between November and December. Chemical composition was analyzed by FoodScan apparatus. The gas chromatography (GC) method was used to determine the content of fatty acids. Results obtained in the presented study indicate that the chemical composition, content of fatty acids *trans* isomers, and lipid quality indices varied between summer and winter cheeses. The summer cheeses were richer sources of MUFA and PUFA compared to winter cheeses. Summer cheeses were also characterized by lower content of SFA, higher content *n* − 3, lower *n* − 6/*n* − 3 ratio, and higher content of DFA. Higher contents of CLA and *trans* C18:1 and C18:2 were found in summer cheeses.

## 1. Introduction

The production of cheese and cottage cheese is the oldest milk processing method mastered by man and is the foundation for the development of the modern dairy industry. The cheese production in Poland increased over the last decade and was accompanied by an enlargement of cheese variety. Polish cheese producers have introduced many new types of cheese to the market that have not been manufactured in the country so far, which clearly shows changes in consumption patterns and consumer preferences [1,2]. In Poland, 3134.4 million liters of drinking milk were produced between January and November 2020, which was 6.9% more than in the same period of 2019. However, during the 11 months of 2020, 310.1 thousand tons of rennet ripened cheese were produced, which was 2.6% less than in 2019 [3]. The nutritional value of cheeses depends mostly on milk characteristics and the production technology. The technological parameters used in cheese production determine the unique and distinct nutritional properties of each cheese type. Substantial amounts of important nutrients, including proteins, bioactive peptides, amino acids, fat, fatty acids, vitamins, and minerals can be found in cheeses [4]. Milk fat is considered to be the most complex fat in the human diet as there are more than 400 distinct fatty acids (FA) [5]. SFAs are the predominant class of fatty acids in milk fat. This group of fatty acids includes: short-chain fatty acids (SCFA), medium and long-chain fatty acids, as well as odd fatty acids (OCFA) and branched-chain fatty acids (BCFA, *iso-* and *anteiso*) [5,6]. Dietary fatty acids may have different effects on human health [7,8]. Saturated fatty acids (SFA), mainly lauric (C12:0), myristic (C14:0), and palmitic (C16:0) acids, may show adverse effects [9,10,11]. Branched-chain fatty acids have anti-tumor effects and improve pancreatic β-cell function [12,13]. BCFA originate from the cell membranes of rumen bacteria that dairy and meat products from ruminants contain and those acids are a unique source of these fatty acids [14]. Monounsaturated fatty acids (MUFA) and polyunsaturated fatty acids (PUFA) are biologically beneficial to human health [10,15]. The best source of natural *trans* fatty acids, such as: vaccenic acid (*trans*11 C18:1, VA) and conjugated linoleic acid (*cis*9*trans*11 C18:2; CLA) in milk fat, which exhibit favorable properties compared to artificial *trans* fatty acids in partially hydrogenated oils [16]. *Trans*11 acid has anti-tumor and anti-atherosclerotic effects [17]. According to literature data [18,19,20,21,22] *cis*9*trans*11 C18:2 acid has a number of health-promoting properties, including anti-carcinogenic, antiatherosclerotic, antioxidant and anti-inflammatory effects. Anti-cancer and anti-atherosclerotic properties are attributed to *cis*9 C18:1 acid and *cis*9*cis*12*cis*15 C18:3 acid [10,23]. *n* − 3 PUFAs prevent heart disease and improve the immune response. The consumption of *n* − 6 and *n* − 3 acids has a positive effect on human health [15]. Milk fat is also a rich source of butyric acid (C4:0), which has beneficial effects on the gastrointestinal tract and supports treatments of chronic diseases of the gastrointestinal tract associated with inflammation [24,25].

The quantitative composition of fatty acids in milk fat changes under the influence of various factors, such as: animal feeding, breed, lactation stage, individual characteristics, climatic conditions, health, age, and others [26,27,28,29,30]. Of the factors mentioned, animal feeding has the greatest influence, which includes pasture feeding, consumption of grass, hay, or silage [31,32,33,34,35,36,37,38]. Fat from milk obtained in the green feed season contains much more C18 acids, mainly C18:1 acid, and much less palmitic and myristic acid than the one obtained in the cowshed feeding season [39,40]. Milk fat from the pasture period of cow’s feeding is also characterized by a higher content of *cis*9*trans*11 C18:2 conjugated linoleic acid and C18:1 and C18:2 *trans* isomers [30,34,38]. A study conducted by Żegarska et al. [38] indicated that the proportion of *cis*9*trans*11 C18:2 acid in fat from the winter period ranged between 0.32% to 0.52% of the total fatty acids composition, whereas in fat from the summer period, it was from 1.06% to 1.76%. The chemical composition and fatty acid composition of cheese may differ from that of milk. Cheese composition depends on the milk’s microbiological and chemical composition, the cheesemaking technology, ripening time, and cheese factory conditions [41,42,43]. Since milk fat and protein are main constituents of cheese, product quality is heavily influenced by their concentrations in milk. The main component determining the quality of milk used for cheese production is the protein content. Milk high in protein, especially casein, results in high yield and good quality cheese products. The fat content is important as well, but especially the ratio of protein or casein content to it is of significance. The quality of cheese, its caloric and nutritional value, physical properties, and chemical composition depends, to a large extent, on the fat content. The fat content in milk can vary within wide limits, from 2.5% to 5% or sometimes even more. The content of protein in the milk varies to a lesser extent in comparison to milk fat content. The biggest fraction within protein content is covered by casein, with is about 3% of all proteins found in milk [44]. Changes in the concentrations of protein and fat in milk significantly influence the composition of the cheese and its yield. Milk protein and fat contents vary greatly according to species, breed, season, health status, stage of lactation, and animal diet [10,33,45]. The type of starter culture used might modify the total content of protein, fat, ash, and fatty acids profile of the cheese, due to the different activity and specificity of proteolytic and lipolytic enzymes [46]. Because of dehydration, protein and fat contents increase during ripening time [4,46,47]. According to literature data [48,49,50,51,52,53], the profile of fatty acids, including the content of CLA in cheeses and fermented drinks, may be influenced by the conditions used in technological processes, the additives used, the activity of the added starter cultures, and ripening time. Rutkowska et al. [54], Zeppa et al. [55], and Serrapica et al. [56] demonstrated that cheeses from summer season had higher unsaturated fatty acids (UFA) content and lower content of saturated fatty acids (SFA) compared to cheeses from winter season.

The aim of the study was to compare and demonstrate whether commercial rennet ripening of cheeses available on the market in summer and winter differ in their chemical composition, fatty acid profile, content of *cis*9*trans*11 C18:2 (CLA) acid, and other *trans* isomers of C18:1 and C18:2 acid and whether they are characterized by different values of lipid quality assessment indices.

## 2. Materials and Methods

### 2.1. Cheeses Samples

To make cheese, you need milk, rennet, microorganisms, and salt. The main stages of the technological process of producing rennet ripened cheeses are: pasteurization and normalization of milk, addition of leaven from pure cultures and calcium chloride, mixing and heating, rennet addition, coagulation and cutting, shaping and pressing of cheese, salting in brine, dripping, and maturation [57]. The quantitative and qualitative diversification of individual ingredients, followed by modifications to the stages of the production process, have led to the development of many types of cheese. Commercial rennet ripened cheeses (Gouda, Edamski, Morski, Edam, Kasztelański, Podlaski) produced from cow’s milk available in the Polish market were used as experimental material. Taking into account the cheese production process, including the ripening time and the shelf-life specified by the manufacturer on the packaging, the analyzed cheeses were divided into two batches. The first batch (labeled as winter cheeses; n = 20) contained cheeses produced in winter and purchased from the market between May and June 2020. The second batch, the same cheeses from the same producers (labeled as summer cheeses; n = 20), contained cheeses produced in summer and purchased from the market between November and December 2020. The products were bought from stores in Olsztyn, Poland. All samples were analyzed in duplicate. 

### 2.2. Analytical Methods 

#### 2.2.1. Chemical Composition

Concentration of fat, protein, water and dry matter in cheese samples were analyzed by FoodScan apparatus (Foss, Hilleroed, Denmark).

#### 2.2.2. Lipid Extraction

Folch’s method was used for fat extraction from cheeses [58].

#### 2.2.3. Preparation of Fatty Acid Methyl Esters

Methyl esters were prepared using the IDF method (ISO 15884:2002) [59]. To each fat sample, N-hexane and 2 M KOH in methanol were added and the mixtures were shaken. In the next step, sodium hydrogen sulphate (NaHSO_4_) was added and the mixtures were centrifuged (3000 min^−1^). The top layer of the prepared methyl esters was collected for chromatographic analysis.

#### 2.2.4. Gas Chromatography (GC) Analysis

The methyl esters were analyzed by the GC method. Chromatographic separation was performed using Hewlett Packard 6890 gas chromatography (Münster, Germany) with a flame ionization detector (FID) and 100 m capillary column (Chrompack, Middelburg, The Netherlands) and internal diameter 0.25 mm. The liquid phase was CP Sil 88 and film thickness was 0.20 μm. The analysis was carried out in the following conditions: column temperature from 60 °C (for 1 min) to 180 °C (Δt = 5 °C/min), detector temperature 250 °C, injector temperature was 225 °C, helium was carrier gas (gas flow 1.5 mL/min). Sample injection volume was 0.4 μL (split mode 50:1). Identification of fatty acids was carried out based on the comparison of their retention time with the retention time of methyl esters of fatty acids of reference milk fat (BCR Reference Materials) of CRM 164 symbol and literature data [60,61,62,63]. The positional *trans* isomers of C18:1 were identified using the standards of methyl esters for these isomers (Sigma-Aldrich, St. Louis, MO, USA and Supelco, Bellefonte, PA, USA). The ratio of their peak area to the total area of all identified acids (% mass fraction) helped with calculations of the proportions of the individual acids.

#### 2.2.5. The Lipid Quality Indices

The following formulas were used to determine Lipid Quality Indices on the basis of fatty acid composition: 

Index of Atherogenicity (AI)—according to Ulbricht and Southgate [11] and Osmari et al. [64]:AI = (C12:0 + (4 × C14:0) + C16:0)/(*n* − 3 PUFA + *n* − 6 PUFA + MUFA)(1)

Index of Thrombogenicity (TI)—according to Ulbricht and Southgate [11] and Osmari et al. [64]:TI = (C14:0 + C16:0 + C18:0)/((0.5 × C18:1) + (0.5 × sum of other MUFA) + 0.5 × *n* − 6 PUFA) + (3 × *n* − 3 PUFA) + *n* − 3 PUFA/*n* − 6 PUFA))(2)

Hypocholesterolaemic fatty acids (DFA)—according to Medeiros et al. [65]: DFA = UFA + C18:0(3)

Hypercholesterolaemic fatty acids (OFA): OFA = C12:0 + C14:0 + C16:0(4)

Hypocholesterolaemic/hypercholesterolaemic ratio (HH)—according to Ivanova and Hadzhinikolova [66]:H/H = (C18:1*n* − 9 + C18:2*n* − 6 + C18:3*n* − 3)/(C12:0 + C14:0 + C16:0)(5)

#### 2.2.6. Assessment of the Coverage of Daily Requirements for Selected Nutrients According to National Nutrition Standards

Based on per capita cheese intake data compiled by the CSO in 2019 and data on fat intake estimated based on the Reference Intake Value (RI) and protein intake based on the recommended Recommended Daily Allowance (RDA) intake for selected population groups presented in the Nutrition Standards for the Polish population by the Institute of Food and Nutrition, the % coverage of the daily requirement for these nutrients was calculated [67,68].

### 2.3. Statistical Analysis 

STATISTICA ver. 13.1 software (Statsoft, Kraków, Poland) was applied to perform the statistical analysis [69]. One-way analysis of variance (ANOVA) with the Duncan’s test was used to find means that were significantly different from each other. Statistically significant differences were accepted at *p* < 0.05.

## 3. Results and Discussion

### 3.1. Chemical Composition

The mean content of fat and water for summer and winter cheeses was observed at a similar level (Figure 1). Winter cheeses were characterized by significantly higher (*p* < 0.05) content of protein than summer cheeses. The composition and quality of cheese are influenced by various factors: the microbiological and chemical composition of milk, the technology of cheese making, the time and conditions of maturation [70,71,72]. Milk protein and fat content can vary due to different species, breed, season, health status, stage of lactation, and animal diet [4,47,73]. In addition, the type of starter culture used in cheese manufacture might modify the total content of protein, fat, ash, and fatty acids profile due to different activity and specificity of proteolytic and lipolytic enzymes [46,74]. The protein and fat content increase during ripening time, which can be caused by partial evaporation of water [4,46].

Protein is an essential macronutrient in the human diet. It is recognised as a key dietary component of human body’s requirements considering its complex metabolic transformations [67]. This is due to the fact that protein metabolism is determined by many factors. Proteins are necessary structural and functional components of every cell in the human body, undergoing energy metabolism as well as intensive interactions with other nutrients that are supplied with food. Moreover, these molecules are essential for the development and growth processes of young organisms, also acting as regulators of gene expression [67]. Proteins are important nutrient because they provide protein nitrogen and specific types of amino acids [67]. Natural sources of proteins and amino acids are raw materials or food products, of which animal products such as dairy (milk, cheese, eggs) and meat, including fish and poultry (except connective tissue proteins low in tryptophan), are complete sources of protein. The main factors influencing human protein requirements are body energy balance, physiological status, age, health status, body weight, physical activity [67,75,76,77]. In the 2017 Nutrition Standards for the Polish population [46], reference intake ranges (RI) were defined for fat and carbohydrates. In contrast, standards for protein and other nutrients were set at levels of average intake EAR and recommended intake RDA, or adequate intake AI. The RDA is the level of nutrient intake that covers the needs of almost all people in a given group. The reference intake (reference macronutrient intake ranges, RI) is the level of macronutrient intake expressed as a percentage of energy requirements. It indicates what range of the percentage of energy from a given macronutrient ensures the maintenance of good health and is associated with a low risk of developing selected chronic diseases [67]. The RDA level (representing the needs of 97–98% of the population, calculated as EAR + 2SD) is intended for intake planning or population-level nutritional goal planning. The recommended daily allowance (RDA) for protein is the amount, based on nitrogen balance (physiological), needed to meet basic nutritional (vital) needs for this component. For adults, it is 0.9 g/kg body weight/day; however, people over 65 years of age are currently advised to consume at least 1 g/kg body weight of protein and to increase their intake in the presence of symptoms of malnutrition or chronic disease to at least 1.2 g/kg body weight (and in justified cases of disease even to 1.5 g/kg body weight/day).

For people with increased physical activity, athletes, an intake of 1.4–2 g/kg body weight/day is recommended. In a sense, this is the minimum amount that needs to be consumed each day in order to keep the body’s immunity at the right level and not get sick. It is assumed that the average of 7 or 10 days, should be within the range of the established daily norm. The correct fulfilment of a person’s protein requirements does not consist of consuming the required amount of protein each day. The average protein intake should not be less than the level of the EAR standard or the lower limit of % energy from protein, which means that it will then be in accordance with the recommendations of Jarosz et al. [67]. In Poland in 2019, the monthly consumption of cheese per urban resident was 1.0 kg, while per rural resident, it was 0.8 kg [68]. Taking into account the data on cheese consumption per capita compiled by the CSO in 2019 and the protein intake data based on the recommended RDA intake for selected population groups presented in the Nutrition Standards for the Polish population by the Institute of Food and Nutrition and the results of protein determinations in the studied cheese samples, the % coverage of the daily requirement for this component was calculated. It was found that the indicated level of intake covered the recommended protein intake in 15.56% for women over 19 years of age and in 13.33% for men over 19 years of age when consuming winter cheeses (Table 1) and in 14.44% for women over 19 years of age and in 12.22% for men over 19 years of age when consuming summer cheeses, whereas for pregnant and lactating women from 8.97% to 10.83% and from 9.66% to 11.67%, respectively.

Table 2 shows the estimated fat intake per capita in Poland based on cheese consumption data in 2019 [54]. Fat intake was estimated based on the Reference Intake (RI) presented in the Nutrition Standards for the Polish Population by the Food and Nutrition Institute [67]. The fat intake standards were developed based on the recommendations of Polish scientific societies and updated opinions of EFSA and FAO/WHO [67]. In developing the standards, the assumption was made that 9 kcal corresponds to one gram of fat [78]. Considering the physiological functions of fats, their insufficient dietary supply may lead to an increased risk of deficiency of fat-soluble vitamins (A, D, E, K), especially among vegetarians [79,80] and central nervous system (CNS) dysfunction, since fat is an important component of this system (including the brain) and fatty acids are the most key components of this organ. Considering the reference intake value (RI), the cheeses studied can be an important source of fat in the daily diet. Data on dietary intake of cheeses in the population in Poland may cover up to about 12% of the reference intake value for fat in the group of women over 19 years of age, while in the group of men with a higher fat requirement, from 7.76 to 8.53% of the RI. In the diets of pregnant and lactating women, whose fat requirements are significant, cheeses provide varying degrees of RI coverage from 9.50% to 11.15% for summer cheeses and from 9.71% to 11.41% for winter cheeses.

### 3.2. Fatty Acids Profile and Lipid Quality Indices in Cheese Fat

In the presented study, saturated fatty acids (SFA) predominated in all examined cheeses. A significantly lower (*p* < 0.05) content of these acids (58.61 ± 1.12%) was found in cheeses from the summer period than in these from the winter period (62.30 ± 0.84%). In addition, all cheese samples contained major SFA in the form of palmitic acid, myristic acid, and stearic acid (Table 3).

Furthermore, monounsaturated fatty acids (MUFA) and polyunsaturated fatty acids (PUFA) content was significantly higher (*p* < 0.05) in summer than winter cheeses (Table 4). In both groups, oleic acid (*cis*9 C18:1) was the major MUFA (Table 3). Whereas linoleic acid (C18:2) and linolenic acid (C18:3) were found to be the major PUFA (Table 3). Short-chain fatty acids (SCFA) content did not differ significantly between summer and winter cheeses (Table 4). The summer cheeses had significantly higher (*p* < 0.05) contents of branched-chain fatty acids (BCFA) compared to winter cheeses. The contents of odd-chain fatty acids (OCFA) in the analyzed cheeses were at a similar level (Table 4). Research conducted by Prandini et al. [81] on cheeses made from cow’s milk indicated that SFA content ranged from 65.23% to 68.52%, MUFA content varied from 27.90% to 31.19%, and PUFA content ranged from 3.48% to 4.17%. Similar results were obtained by Paszczyk and Łuczyńska [82], who demonstrated that content of MUFA and PUFA in commercial cheeses, made from cow’s milk, purchased from September to December was 27.92% and 3.31%, respectively. 

In the presented study, *n* − 3 PUFA content was significantly higher (*p* < 0.05) in summer cheeses than winter cheeses, whereas the content of *n* − 6 PUFA was at similar level (Table 4). The values of lipid quality indexes varied between summer and winter cheeses (Table 4). Summer cheeses contained significantly more (*p* < 0.05) desirable hypocholesterolemic fatty acids (DFAs) and significantly less (*p* < 0.05) hypercholesterolemic fatty acids (OFA). Furthermore, the AI and TI indexes were significantly higher (*p* < 0.05) in winter cheeses. The AI and TI indexes are believed to have an impact on the risk of cardiovascular diseases [11]. The higher values of these indices higher risk of certain diseases. Dairy products with lower AI values can reduce levels of total cholesterol and LDL-cholesterol in human plasma [66]. A significantly higher (*p* < 0.05) H/H content was found in summer cheeses than in winter cheeses. Moreover, summer cheeses had more favorable ratio of *n* − 6 to *n* − 3 acids (3.04) compared to winter cheeses (4.52), which may indicate better health-promoting properties of summer cheeses. Proportions of fatty acids specific groups in dairy products are of special importance from a nutritional perspective. Excessive amounts of *n* − 6 PUFA and very high *n* − 6/*n* − 3 ratio, which can be typically found in today’s Western diets, promote the pathogenesis of many diseases, whereas increased levels of *n* − 3 PUFA (a low *n* − 6/*n* − 3 ratio) exert suppressive effects [83,84,85]. Similar results were obtained by Hirigoyen et al. for the *n* − 6/*n* − 3 ratio in the cheeses [86]. The authors demonstrated that this ratio in “Colonia” cheeses produced from cow’s milk in spring was 4.47 and 3.29 in cheeses produced in autumn.

### 3.3. The Content of CLA and Trans C18:1 and Trans C18:2 Fatty Acids in Cheese

The content of *cis*9*trans*11 C18:2 (CLA) and total content of C18:1 and C18:2 *trans* isomers in summer and winter cheeses are presented on Figure 2. The conducted analyses indicated that summer cheeses had significantly higher (*p* < 0.05) content of CLA than winter cheeses (0.88% and 0.44%, respectively). 

Dairy products, such as milk, cheese, and yogurt are the principal natural sources of CLA in the human diet. The term conjugated linoleic acid (CLA) refers to a group of positional and geometric isomers of linoleic acid characterized by the presence of two conjugated double bonds. The main isomer in dairy products is rumenic acid, *cis*9*trans*11 CLA, which represents 75 to over 90% of total CLA in milk fat [87]. CLA is an intermediate in the biohydrogenation of linoleic acid in the rumen, but its main source is endogenous synthesis from *trans* 11 C18:1 acid by Δ9-desaturase [88]. According to Żegarska et al. [89], the content of CLA in milk fat ranged from 0.32% to 0.52% in the winter season and from 1.06% to 1.76% in the pasture period. Several studies have also demonstrated a large variation of CLA content in cheese. Fritsche and Steinhart [90] analyzed German cheese made from cow’s milk, and found that CLA content in total fatty acid composition varied from 0.40% to 1.70%. Donmez et al. [91] indicated that CLA content in fat from Turkish cheeses ranged from 0.44 to 1.04 g/100 g. According to Grega et al. [92], CLA content in commercial cheeses ranged from 0.20% to 0.95% in the winter season and from 0.61% to 1.57% in the summer season. Żegarska et al. [89] observed that in hard commercial cheeses, purchased in February and March, CLA content in total fatty acid composition ranged from 0.48% to 1.68%, while cheeses bought in October and November had from 0.97% to 1.46%. In commercial cheeses bought in the period from November to December analyzed by Paszczyk and Łuczyńska [82], the CLA content varied from 0.46% to 0.85% of total fatty acids.

In the group of *trans* C18:1 isomers, the *trans* 11 isomer is the dominant one; it accounts for about 40–50% in total C18:1 *trans* fatty acids [93]. The animal feeding system can influence the content of this isomer in milk fat [94,95]. In the presented study, the total content of C18:1 and C18:2 *trans* isomers in summer cheeses was significantly higher (*p* < 0.05) compared to winter cheeses (Figure 2). Obtained results are in consistence with Żegarska et al. [38], who demonstrated that *trans* C18:1 isomers ranged from 3.57% to 5.37%, and from 1.26% to 1.84% in summer and winter milk, respectively. Moreover, research conducted on commercial cheeses by Żegarska et al. [89], purchased in February and March, also demonstrated that total content of *trans* C18:1 isomers ranged from 1.65% to 4.42% in total fatty acids composition. Whereas, the proportion of *trans* C18:1 isomers in cheeses purchase from October and November varied from 4.14% to 4.69%. Authors also showed that content of *trans* C18:2 isomers in cheeses purchased from February to March ranged from 0.44% to 1.17%, and from 0.96% to 1.11% in cheeses bought from October to November, which is consistent with the presented study. In addition, Paszczyk and Łuczyńska [82] indicated that total content of C18:1 and C18:2 *trans* isomers, in Polish cheeses made from cow’s milk (purchased from September to December), ranged from 2.58% to 3.27% and *trans* C18:2 ranged from 0.50% to 0.75% in total fatty acids composition.

## 4. Conclusions

Interest in cheeses, and the demands of increasingly aware consumers mean that new varieties of cheese are constantly being supplied to the market. Consumers are paying more and more attention to the composition of the food they eat, while at the same time looking for products that are as natural as possible. Animal feeding influences the quality and chemical composition of milk, and thus the quality and composition of cheeses. Therefore the evaluation of cheese properties is becoming more and more important. According to the obtained results, the chemical composition, content of fatty acids, CLA, other *trans* isomers, and lipid quality indices varied in summer and winter cheeses. Winter cheeses were characterized as a better source of protein than summer cheeses and might be a better choice, meeting daily protein requirements of different population groups.

## Figures and Tables

**Figure 1 animals-12-00198-f001:**
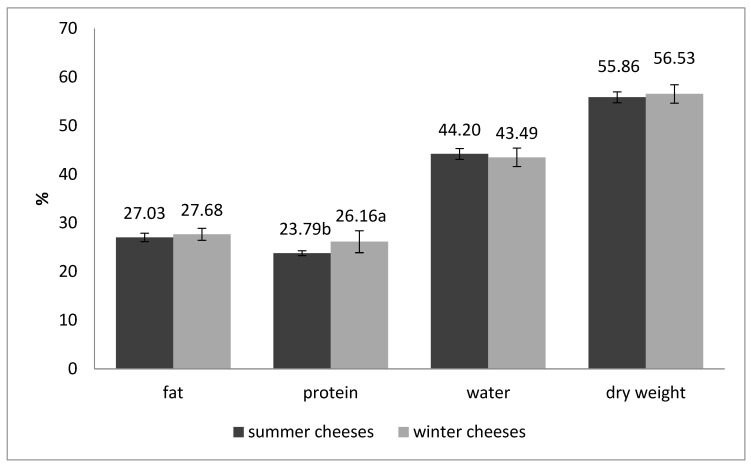
The chemical composition (%) of summer and winter cheeses made from cow’s milk. a,b—values differ significantly between groups (*p* < 0.05).

**Figure 2 animals-12-00198-f002:**
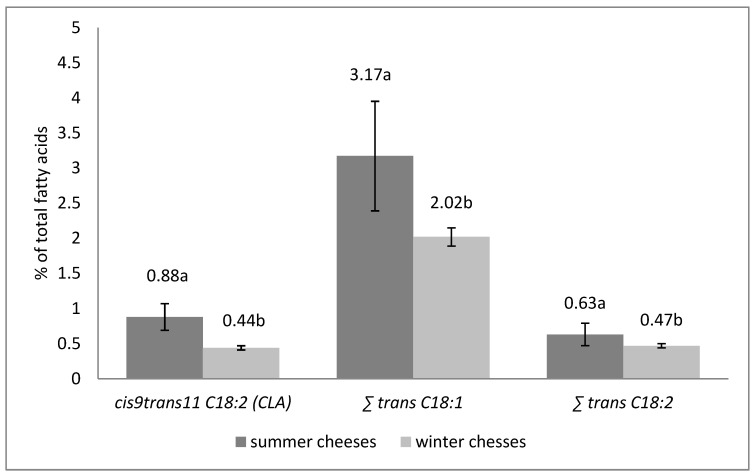
The content of *cis*9*trans*11 C18:2 (CLA) and total C18:1 and C18:2 *trans* isomers (% of total fatty acids) in summer and winter cheeses made from cow’s milk. ∑*trans* C18:1—sum of *trans* isomers of C18:1, ∑ sum of *trans* C18:2 without *c*9*t*11 C18:2, a,b—statistically significant differences (*p* < 0.05).

**Table 1 animals-12-00198-t001:** Covering the daily protein requirements after eating a portion of cheese.

Group/Age	Body Weight [kg]	Reference Protein [g/kg b.w./Day]	National Food Ration Protein -RDA [g/kg b.w./Day]	Cheese Consumption per Capita in 2019 * [kg Cheese/Day]	DDC [%]	
					Summer Cheeses	Winter Cheeses
Women ≥ 19	45–75	0.80	0.90	0.033	14.44	15.56
Men ≥ 19	55–85	0.80	0.90		12.22	13.33
Pregnat women ˂ 19 age	45–75	1.10	1.20		10.83	11.67
Pregnat women ≥ 19 age	45–75	1.10	1.20		10.83	11.67
Breast-feeding women ˂ 19 age	45–75	1.3	1.45		8.97	9.66
Breastfeeding women ≥ 19 age	45–75	1.3	1.45		8,97	9.66

* Data in accordance with the Central Statistical Office in Poland in 2019 [68]. Source: own study based on Jarosz et al. [67]; Abbreviations: RDA—Recommended Daily Allowance; DDC—Daily Demandes Coverage.

**Table 2 animals-12-00198-t002:** Covering the daily fat requirements after eating a portion of cheese.

Group/Age		Body Weight [kg]	RI [g/Person/Day]	Cheese Consumption per capita in 2019 * [kg Cheese/Day]	DDC [%]	
					Summer Cheeses	Winter Cheeses
**Women ≥ 19**			77	0.033	11.58	11.86
		55	85		10.49	10.74
**Men ≥ 19**		65	107		8.34	8.53
			115		7.76	7.94
**Pregnat women**	**I**	65	80		11.15	11.41
	**II**		87		10.25	10.49
	**III**		93		9.60	9.82
**Breastfeeding women**		85	94		9.49	9.71

* Data in accordance with the Central Statistical Office in Poland in 2019 [68]. Source: own study based on Jarosz et al. [67]; Abbreviations: RI—Reference Intake; DDC—Daily Demandes Coverage.

**Table 3 animals-12-00198-t003:** Mean ± SD and range of fatty acids composition (% of total fatty acids) in summer and winter cheeses made from cow’s milk.

Summer Cheeses				Winter Cheeses		
	Mean ± SD		Min–Max	Mean ± SD		Min–Max
*n*	20			20		
C4:0	3.05	±0.36	2.56–3.80	2.80	±0.36	2.09–3.25
C6:0	2.12	±0.08	2.00–2.25	1.98	±0.27	1.54–2.25
C8:0	1.35	±0.06	1.24–1.44	1.33	±0.06	1.25–1.41
C10:0	3.07	±0.17	2.81–3.29	3.18	±0.08	3.09–3.32
C10:1	0.33	±0.02	0.30–0.36	0.33	±0.01	0.31–0.34
C11:0	0.06	±0.01	0.04–0.07	0.06	±0.01	0.5–0.07
C12:0	3.55	±0.16 ^b^	3.22–3.69	3.75	±0.13 ^a^	3.50–3.88
C12:1	0.08	±0.01	0.07–0.09	0.09	±0.00	0.08–0.09
C13:0 *iso*	0.10	±0.01	0.09–0.11	0.10	±0.01	0.09–0.11
C13:0	0.13	±0.02	0.10–0.17	0.12	±0.01	0.10–0.14
C14:0 *iso*	0.13	±0.01	0.11–0.14	0.13	±0.01	0.12–0.15
C14:0	11.31	±0.24 ^b^	11.18–11.85	12.10	±0.23 ^a^	11.66–12.45
C15:0 *iso*	0.28	±0.02	0.23–0.31	0.26	±0.02	0.23–0.29
C15:0 *aiso*	0.56	±0.06 ^a^	0.48–0.63	0.51	±0.02 ^b^	0.49–0.53
C14:1	1.03	±0.13	0.85–1.16	1.09	±0.04	1.02–1.12
C15:0	1.26	±0.06	1.18–1.34	1.22	±0.04	1.17–1.27
C16:0 *iso*	0.31	±0.03	0.28–0.36	0.31	±0.03	0.26–0.34
C16:0	29.09	±1.77 ^b^	26.90–31.75	32.53	±0.40 ^a^	32.22–32.64
C17:0 *iso*	0.44	±0.04 ^a^	0.36–0.49	0.36	±0.02 ^b^	0.33–0.39
C17:0 *aiso*	0.23	±0.03 ^a^	0.19–0.26	0.18	±0.00 ^b^	0.17–0.19
C16:1	1.65	±0.25	1.24–2.02	1.65	±0.13	1.57–1.97
C17:0	0.86	±0.38 ^a^	0.68–1.92	0.72	±0.01 ^b^	0.71–0.74
C17:1	0.25	±0.02	0.23–0.29	0.25	±0.01	0.24–0.26
C18:0	9.95	±0.53	9.20–10.72	9.64	±0.48	8.89–10.26
*t*6 − *t*9 C18:1	0.44	±0.04	0.41–0.52	0.42	±0.02	0.37–0.34
*t*10 + *t*11 C18:1	2.20	±0.65 ^a^	1.27–3.05	1.24	±0.09 ^b^	1.13–1.35
*t*12 C18:1	0.30	±0.04	0.26–0.38	0.28	±0.02	0.24–0.30
*c*9 C18:1	19.59	±0.57 ^a^	18.83–20.60	19.04	±0.43 ^b^	18.54–19.70
*c*11 C18:1	0.61	±0.03	0.57–0.67	0.61	±0.04	0.56–0.64
*c*12 C18:1	0.24	±0.04 ^b^	0.19–0.30	0.27	±0.03 ^a^	0.21–0.33
*c*13 C18:1	0.09	±0.01	0.07–0.10	0.09	±0.01	0.07–0.10
*t*16 C18:1	0.34	±0.05 ^a^	0.27–0.42	0.29	±0.02 ^b^	0.25–0.31
C19:0	0.19	±0.02 ^a^	0.14–0.23	0.16	±0.02 ^b^	0.14–0.18
*c*9 *t*13 C18:2	0.20	±0.03 ^a^	0.15–0.25	0.16	±0.01 ^b^	0.15–0.17
c9 t12 C18:2	0.18	±0.03 ^a^	0.14–0.22	0.15	±0.01 ^b^	0.13–0.17
*t*11 *c*15 C18:2	0.26	±0.10 ^a^	0.10–0.37	0.10	±0.02 ^b^	0.06–0.14
*c*9 *c*12 C18:2	1.52	±0.19 ^a^	1.28–1.78	1.54	±0.14 ^a^	1.39–1. 86
C20:0	0.15	±0.01	0.14–0.16	0.15	±0.01	0.13–0.16
C20:1	0.11	±0.01	0.10–0.12	0.11	±0.00	0.11–0.12
*c*9*c*12*c*15 C18:3	0.53	±0.13 ^a^	0.31–0.66	0.35	±0.07 ^b^	0.26–0.48
*c*9*t*11 C18:2 (CLA)	0.88	±0.19 ^a^	0.51–1.12	0.44	±0.03 ^b^	0.39–0.48

*n*—number of samples, Min—minimum value, Max—maximum value, Mean—mean value, SD—standard deviation, ^a,b^—statistically significant differences (*p* < 0.05).

**Table 4 animals-12-00198-t004:** Mean ± SD of sum of fatty acids (% of total fatty acids) and nutritional indices in summer and winter cheeses made from cow’s milk.

Fatty Acids	Summer Cheeses		Winter Cheeses	
*n*	20		20	
	Mean ± SD	Min–Max	Mean ± SD	Min–Max
ΣSCFA ^1^	9.58 ± 0.38	9.10–10.19	9.29 ± 0.69	8.08–10.07
ΣBCFA ^2^	2.08 ± 0.18 ^a^	1.76–2.24	1.84 ± 0.06 ^b^	1.76–1.97
ΣOCFA ^3^	2.50 ± 0.45	2.18–2.57	2.21 ± 0.04	2.15–2.27
ΣSFA ^4^	58.61 ± 1.12 ^b^	57.24–60.41	62.30 ± 0.84 ^a^	61.40–63.99
ΣMUFA ^5^	27.25 ± 0.60 ^a^	26.32–28.62	25.75 ± 0.50 ^b^	25.19–26.48
ΣPUFA ^6^	3.68 ± 0.60 ^a^	2.83–4.51	2.74 ± 0.14 ^b^	2.61–3.00
*n* − 3	0.53 ± 0.13 ^a^	0.31–0.66	0.35 ± 0.05 ^b^	0.31–0.42
*n* − 6	1.52 ± 0.19	1.28–1.78	1.54 ± 0.14	1.39–1.86
*n* − *6/n* − *3*	3.04 ± 0.81 ^b^	2.42–4.74	4.52 ± 0.99 ^a^	3.56–5.64
UFA ^7^	30.93 ± 0.94 ^a^	29.29–32.44	28.50 ± 0.47 ^b^	27.75–29.16
DFA ^8^	40.89 ± 1.38 ^a^	38.49–42.58	38.14 ± 0.91 ^b^	36.86–39.42
OFA^9^	48.66 ± 1.62 ^b^	46.68–51.21	52.60 ± 1.15 ^a^	51.14–55.10
AI ^10^	2.01 ± 0.07 ^b^	1.85–2.12	3.06 ± 0.05 ^a^	3.00–3.17
TI ^11^	1.99 ± 0.11 ^b^	1.83–2.14	3.79 ± 0.06 ^a^	3.71–3.87
H/H ^12^	0.49 ± 0.03 ^a^	0.45–0.53	0.43 ± 0.01 ^b^	0.42–0.45

*n*—number of samples; Mean—mean value; SD—standard deviation; Min—minimum value; Max—maximum value; ^a,b^—values denoted in rows by different letters indicate statistically significant differences (*p* < 0.05); ^1^ ΣSCFA: sum of short-chain fatty acids (C4:0–C10:0); ^2^ ΣBCFA—all branched-chain fatty acids; ^3^ ΣOCFA—all odd-chain fatty acids; ^4^ SFA—all saturated fatty acids (without SCFA); ^5^ ΣMUFA—sum of monounsaturated fatty acids; ^6^ ΣPUFA—sum of polyunsaturated fatty acids; ^7^ UFA—sum of unsaturated fatty acids (ΣMUFA + ΣPUFA); ^8^ DFA—hypocholesterolemic fatty acids (ΣUFA + C18:0); ^9^ OFA—hypercholesterolemic fatty acids (ΣSFA-C18:0); ^10^ AI—(Index of Atherogenicity); ^11^ TI—(Index of Thrombogenicity); ^12^ H/H—(hypocholesterolaemic/hypercholesterolaemic ratio).

## Data Availability

Data is contained within the article.

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
