# Peer review of "Chemical Composition, Fatty Acid Profile, and Lipid Quality Indices in Commercial Ripening of Cow Cheeses from Different Seasons"

_animals, 2022, doi:10.3390/ani12020198_

Round 1

Reviewer 1 Report

The Authors of the manuscript animals-1421744 measured the basic nutrients and fatty acids profile in rennet ripening cheeses purchased commercially in Olsztyn (Poland). They examined 40 samples in duplicate. The whole article lacks of novelty. The examination of fatty acids profile in different dairy products, and their seasonal changes resulting from differences in animals feeding, have been thoroughly investigated. The Authors did not include any research hypothesis and did not describe any aspects of novelty in their study. They should explain in details what is the novelty of the study, what made the Authors to perform their study, what is the hypothesis and the expected results of their study? Moreover, my major concern relates to the study design. They claim, that the examined samples were divided in two batches, according to production season. How the production season was established by the Authors? There is neither information on production data nor information on the date of raw material obtaining. Taking into account very superficial characteristic of the examined samples, as well as the possible differences in ripening time, the claiming on production season from purchasing date is not very reliable. In the ‘introduction’ section the Authors focused mainly on fatty acids profile of dairy fat. However, they should mentioned all the most characteristic fatty acids of milk fat, including volatile fatty acids and odd- and branched fatty acids (OBCFA). In ‘results and discussion’ they mainly refer to the claiming how consumption of such ripening cheeses will correspond to the nutritional needs of people in terms of proteins and fats. In my opinion ‘introduction’  and ‘results’ are not consistent. Hence, I cannot confirm, that this manuscript meets the aims and scopes of “animals” journal, which publishes original research articles, reviews and communications that offer substantial new insight into any field of study that involves animals, including zoology, ethnozoology, animal science, animal ethics and animal welfare. In this manuscript no ‘animalistic’ aspects are taken into consideration. The Authors should reflect their manuscript and make it more suitable for ‘Foods’ or ‘Nutrients’. I recommend its rejection from ‘animals’.

Author Response

Answer in the attachment

Reviewer 2 Report

Manuscript ID: animals-1421744

Chemical Composition, Fatty Acid Profile and Lipid Quality Indices in Commercial Cowʹs Ripening Cheeses from Different Seasons

General remarks

Dear authors,

I have revised the abovementioned manuscript. As you can see, I had to add the line numbers (see the attached file), that were missing in your version. Numbering the text’ lines is a fundamental tool for reviewers, so much so that the journal's instructions clearly indicate this. In the file that I have attached, you will find a numbering of the lines for each page (on the pdf format I was unable to insert the continuous numbering).

Leaving aside the unfortunate inconvenience, the research topic is interesting from a scientific point of view and consistent with the journal objectives. Nevertheless, in my opinion, the manuscript needs to be carefully revised in several parts. My major concerns are relative to the materials and methods section, in which I believe important information has been omitted.

My comments are listed below, section by section. Hoping to have contributed to improving the manuscript quality. Good works.

Specific comments

Title: in my opinion, the authors might consider introducing the concept of the survey into the title.

L 7 (and along whit the text): please, replace “cow’s milk” whit “cows’ milk”. Thanks.

Simple summary

L 18: please, replace “our body” whit “our bodies”. In my opinion, I would rephrase the sentence as follows: Cheese and other dairy products are important components of the diet that have a positive effect on human health. Thanks

L 20: please, delete the colon after “such as”. Thanks

L 21: please, add a comma after “age” (…..health, age, and). Thanks.

L 28-29: I believe you meant lipids rather than liquids. Furthermore, I suggest to add “the “ before “content” and a comma after “indices”. Thanks.

Abstract

L 37: please, add the article “Gas chromatography”. Thanks

L 39: please, add the article at “content of fatty acids…”. Thanks

Keywords: in my opinion, the keywords list is appropriate.

Introduction

L 7 (page 2): please, replace “renneted” whit “rennet”. Thanks.

L 8 (and along whit the text): I ask the authors to pay attention to the double spaces in the text. Thanks for your patience.

L 14 (page 2; and along whit the text): please, replace “our health” whit “human health”. Thanks.

L 16 (page 2): please, replace “affects” whit “effects”. Thanks.

L 21 (page 2; and along whit the text): please, add the article when the fatty acids were cited at the start of the sentence. In addition, I would avoid bold to indicate fatty acids. Thanks.

L 31 (page 2): please, add a comma before “and others”. Thanks.

L 32-34 (page 2): in my opinion, the statement “Fat from milk obtained in the green feed season contains much more C18 acids, mainly C18:1 acid, and much less palmitic and myristic acid than the one obtained in the cowshed feeding season” should be supported by one or more references, among which I suggest including https://doi.org/10.3168/jds.2018-14710. Thanks.

L 36 (page 2): please, replace “conduced” with “conducted”. Thanks.

L 47 (page 2): in addition to Rutkowska et al. [42] and Zeppa et al. [43], the issue of seasonal variations of the cheese quality is also well addressed by https://doi.org/10.3390/foods9081091, which I suggest using as a reference, thanks.

L 52 (page 2): authors are invited to specify that they studied commercial cheeses. In addition, I would suggest emphasizing that a survey has been conducted. Thanks.

Materials and methods

L 3-10 (page 3) - Cheeses samples: As stated by the authors, the chemical composition and, consequently, the nutritional characteristics of cheese are strictly related to the quality of the raw milk from which it originates and to the specific technological process used. Since these are commercial cheeses, I realize that the authors may not know the conditions under which the milk was produced. By contrast, I believe it is possible to provide information on the types of cheese sampled, on the specific cheesemaking procedures, and on the aging technique and time. Among other things, it is not clear whether the cheeses that make up the two batches (summer and winter) are the same. If possible, the authors are asked to provide this useful (in my opinion) information. Thanks.

L 16 (page 4, and along whit the study): acronymous can be specified at the first mention (see RDA). Thanks.

Statistical analysis: authors are invited to indicate and accurately describe the statistical model used. Thanks.

Results and discussion

L 27 (page 4): please, replace “obserwed” whit “observed”.

L 31 (page 4): in addition to the De Marchi [56] and Formaggioni [57] studies, I also suggest referring to the work of https://doi.org/10.1111/1471-0307.12640, which extensively deals with the issue of factors influencing the quality and composition of cheese. Thanks.

L 28 (page 5): please, add a space within “are” and “necessary”. Thanks.

L 14 (page 8, and along whit the study): according to the journal’ template, authors are asked to avoid commercial “&”, replacing it whit “and”. Thanks.

L 34 (page 10): a comma isn’t usually needed after conjunction. Thanks.

Conclusions

L 50 (page 10): after an introductory word or phrase, a comma is best. Thanks.

Author Response

Answer in the attachment

Reviewer 3 Report

Revision

Manuscript ID 1421744

Chemical Composition, Fatty Acid Profile and Lipid Quality Indices in Commercial Cowʹs Ripening Cheeses from Different Seasons

The manuscript did not report line numbers

Page 1,  Simple Summary in line 5:  In this context it is correct to use animal feeding. It is advisable to replace animal nutrition with animal feeding

Page1,  in the Abstract line 3:  replace "from different seasons" with "... produced/made in different seasons"

Page1,  in Keywords: It is recommended to add "cow cheeses"

Page 2, in Introduction, line 28: It is advisable to replace animal nutrition with animal feeding

Page 2, in Introduction, line 29-30: “Of the above-mentioned factors, the most important influence is the diet [27-34].”

It is advisable to specify which parameters and/or factors of the animal's diet influence the quantitative and qualitative composition of milk fat

Page 2, Introduction: The title indicates the chemical composition of the cheese as well as the fatty acid profile. In order to provide the reader with a complete panorama of the cheeses, it is suggested to include in the introduction the references relating to the chemical composition and not only for fatty acids.

Page 2, Introduction: In order to have a complete picture of the cheese produced in winter and summer, it would also be interesting to know the quantity of salt present in the cheeses. The salt content is an important parameter for human health and for formulating a balanced diet.

To improve the information of the work, the authors could report the salt content of the cheeses.

Page 3, Materials and Methods, line 1: For a better understanding for the reader, it is recommended to report the characteristics of the commercial cheese used in the study. Give a brief description of production technologies of cheeses and/orthe general characteristics of the cheeses such as type of Curd / Pasta, Starter, Rennet, Curd, Cooking, Salting, etc.

It is suggested to indicate, in Material and Methods, the ripening days of the cheeses used in the experimentation.

Page 5, Results and Discussion line 3:  it is suggested to put a space between are and necessary

Page 9, Results and Discussion,  In paragraph 3.3. The Content of CLA and Trans C18: 1 and Trans C18: 2 Fatty Acids in Cheese

 It is advisable to report the updated references on the synthesis of the CLA.

According to

  • Griinari, J.M. and D.E. Bauman, 1999. Biosynthesis of conjugated linoleic acid and its incorporation into meat and milk in ruminants. In M.P. Yurawecz, M.M. Mossoba, J.K.G. Kramer, M.W. Pariza and G.J. Nelson (ed) Advances in Conjugated Linoleic Acid Research. Vol. I., pp: 180-200. AOCS Press, Champaign, IL.
  • Griinari, J.M., B.A. Corl, S.H. Lacy, Chouinard, P.Y., K.V.V. Nurmela and D.E. Bauman, 2000. Conjugated linoleic acid is synthesized endogenously in lactating dairy cows by DELTA 9-desaturase. J. Nutr., 130: 2285-2291.

CLA is mainly synthesized at the mammary gland level through an endogenous synthesis by Delta 9 desaturase on the TVA precursor.

Page 10, Conclusion,  line 4: 

Replace the word "Animal nutrition" with "Animal feeding". The concept of Animal feeding, in this case, is more appropriate. In this case, the authors closely relate the animal feed intake as grass, pasture, hay etc. with the quality of the product.

References

  • The official language of the journal is English, it is suggested to translate the titles of the articles, not in English, cited in the references and add it to the original title. Translate the title of references 1,2, 3, 21, 30, 32, 42, 53, 54, 65 into English.
  • It is recommended to correct the title of the reference 35

Author Response

Answer in the attachment

Round 2

Reviewer 1 Report

I do appreciate all the changes made by the Authors in their manuscript. Although I did not find any justification of claiming of production season based on the time of purchasing and I do not see strongly pronounced aspects of novelty in the text, I give my recommendation to this manuscript to be published in this Special Issue of "Animals". However, if not the Special Issue, I still find this manuscript more suitable for "Fodds" or "Nutrients".

Reviewer 2 Report

Dear authors,
I have evaluated the revised version of the manuscript (ID: animals-1421744). Given the changes made, I have no doubts. In my opinion, beyond the formal aspects, the revised version of the manuscript can be published in the present form. 
Congratulations and good luck